# Theophylline: Old Drug in a New Light, Application in COVID-19 through Computational Studies

**DOI:** 10.3390/ijms23084167

**Published:** 2022-04-09

**Authors:** Luis M. Montaño, Bettina Sommer, Juan C. Gomez-Verjan, Genaro S. Morales-Paoli, Gema Lizbeth Ramírez-Salinas, Héctor Solís-Chagoyán, Zuly A. Sanchez-Florentino, Eduardo Calixto, Gloria E. Pérez-Figueroa, Rohan Carter, Ruth Jaimez-Melgoza, Bianca S. Romero-Martínez, Edgar Flores-Soto

**Affiliations:** 1Departamento de Farmacología, Facultad de Medicina, Universidad Nacional Autónoma de México, Ciudad de México 04510, CP, Mexico; lmmr@unam.mx (L.M.M.); jaimezruth@hotmail.com (R.J.-M.); biancasromero_@hotmail.com (B.S.R.-M.); 2Laboratorio de Hiperreactividad Bronquial, Instituto Nacional de Enfermedades Respiratorias “Ismael Cosío Villegas”, Ciudad de México 14080, CP, Mexico; bsommerc@hotmail.com; 3Dirección de Investigación, Instituto Nacional de Geriatría, Ciudad de México 10200, CP, Mexico; jver-jan@inger.gob.mx (J.C.G.-V.); mgenas.com@hotmail.com (G.S.M.-P.); 4Laboratorio de Diseño y Desarrollo de Nuevos Fármacos e Innovación Biotecnológica (Laboratory for the Design and Development of New Drugs and Biotechnological Innovation), Escuela Superior de Medicina, Instituto Politécnico Nacional, Plan de San Luis y Díaz Mirón S/N, Col. Santo Tomas, Ciudad de México 11340, CP, Mexico; gemali86@hotmail.com; 5Departamento de Inmunología, Instituto de Investigaciones Biomédicas, Universidad Nacional Autónoma de México, Circuito Escolar s/n, Ciudad de México 14510, CP, Mexico; 6Laboratorio de Neurofarmacología, Instituto Nacional de Psiquiatría “Ramón de la Fuente Muñiz”, Ciudad de México 14370, CP, Mexico; hecsolch@imp.edu.mx (H.S.-C.); zulyarmandosf@gmail.com (Z.A.S.-F.); 7Departamento de Neurobiología, Dirección de Investigación en Neurociencias, Instituto Nacional de Psiquiatría “Ramón de la Fuente Muñiz”, Ciudad de México 14370, CP, Mexico; ecalixto@imp.edu.mx; 8Instituto Nacional de Neurología y Neurocirugía, Unidad Periférica en el Estudio de la Neuroinflamación en Patologías Neurológicas, Ciudad de México 06720, CP, Mexico; gera.pfi3@gmail.com; 9Laboratorio de Investigación en Inmunología y Proteómica, Hospital Infantil de México Federico Gómez, Ciudad de México 06720, CP, Mexico; 10FRACGP/MBBS, Murchison Outreach Service Mount Magnet Western Australia, Mount Magnet, WA 6530, Australia; kingswood71@gmail.com

**Keywords:** theophylline, COVID-19, SARS-CoV-2, immunomodulatory effects, antiviral activity, molecular docking, network pharmacology, quantitative systems pharmacology

## Abstract

Theophylline (3-methyxanthine) is a historically prominent drug used to treat respiratory diseases, alone or in combination with other drugs. The rapid onset of the COVID-19 pandemic urged the development of effective pharmacological treatments to directly attack the development of new variants of the SARS-CoV-2 virus and possess a therapeutical battery of compounds that could improve the current management of the disease worldwide. In this context, theophylline, through bronchodilatory, immunomodulatory, and potentially antiviral mechanisms, is an interesting proposal as an adjuvant in the treatment of COVID-19 patients. Nevertheless, it is essential to understand how this compound could behave against such a disease, not only at a pharmacodynamic but also at a pharmacokinetic level. In this sense, the quickest approach in drug discovery is through different computational methods, either from network pharmacology or from quantitative systems pharmacology approaches. In the present review, we explore the possibility of using theophylline in the treatment of COVID-19 patients since it seems to be a relevant candidate by aiming at several immunological targets involved in the pathophysiology of the disease. Theophylline down-regulates the inflammatory processes activated by SARS-CoV-2 through various mechanisms, and herein, they are discussed by reviewing computational simulation studies and their different applications and effects.

## 1. Introduction

Theophylline (1,3-dimethyl-7H-purine-2,6-dione) is a dimethylxanthine derived from the xanthine purine base and composed of two methyl groups located at positions 1 and 3. Naturally, it is present in black tea (0.02–0.04% dry weight) [1,2] coffee (5 mg/kg) [3], chocolate, dried mate [4], and related foodstuffs [5]. Theophylline is rapidly and completely absorbed in the gastrointestinal tract after oral administration in solution reaching maximum serum levels around 1.5 and 2 h after intake. This alkaloid does not undergo any appreciable pre-systemic elimination, distributes freely into fat-free tissues, and is extensively metabolized in the liver involving at least two cytochrome P450 isozymes [6]. Theophylline serum half-life ranges from about 3 to 12.8 (average 7–9) h [7,8] and can be toxic in doses of 7.5 mg/kg or above [9]. Special consideration in the administration of theophylline must be taken in circumstances that can represent reduced drug clearance, for instance, in patients with liver or pulmonary disease (pneumonia, COPD) and heart failure. In contrast, an increased clearance can be expected in children <16 years and smokers [10]. Additionally, several drug interactions should be considered when theophylline is administrated, for example, decreased clearance in association with erythromycin, quinolones, allopurinol, cimetidine, serotonin uptake inhibitors, and the 5-lipoxygenase inhibitor zileuton, and a higher metabolism when administered in conjunction with phenytoin, phenobarbital, and rifampicin [10]. On the other hand, common side effects observed with theophylline treatment are headaches, gastrointestinal discomfort, insomnia, nausea, vomiting, and at higher doses, serious side effects such as seizures and cardiac arrhythmias, primarily due to antagonism of adenosine 1 receptor (A_1_Rs), might develop. Low-dose, slow-release treatment seems to be well-tolerated in long-term treatments for COPD and asthmatic patients, reducing exacerbations and the probability of serious side effects [10,11].

The methylxanthines theophylline and dyphylline are used in the treatment of airway obstruction caused by clinical conditions such as asthma, infant apnea, chronic bronchitis, emphysema, and chronic obstructive pulmonary disease (COPD) [12,13] and have recently been proposed [14,15] and used as a supplement to treat coronavirus disease 2019 (COVID-19) patients [16,17,18]. Furthermore, recent studies have contributed to establishing theophylline’s great therapeutic potential in the COVID-19 treatment. Wall et al. carried out a retrospective study on COVID-19 patients that required oxygen and received either theophylline or pentoxifylline. Patients with a history of asthma or COPD were given the former, and all other patients received pentoxifylline. Evaluations were done comparing C-reactive protein (CRP) concentrations and ROX score (defined as the ratio of oxygen saturation measured by pulse oximetry/FiO2 to respiratory rate) between a control group constituted by COVID-19 patients receiving standard medication and the group receiving xanthines from day 1 to day 4 of therapy, and results showed an increase in the ROX score (mean: 2.9) and a decrease in CRP (mean: −0.7) and mortality (24%) for the theophylline/pentoxifylline group. Even though it was a non-randomized study, the data obtained point out that this treatment could be associated with benefits for COVID-19 patients and warrants further research [16].

Additionally, Dahiya et al. reported the effects of theophylline or etophylline on sinus bradycardia, one of the most common arrhythmias found in COVID-19 patients. This symptom is possibly related to viral myocarditis, myocardial ischemia or might be a side effect of COVID-19 medication. Ten COVID-19 patients that developed sinus node dysfunction received etophylline or theophylline prolonged-release tablet (150 mg) once a day, and a normal heart rate was monitored 72 h after the treatment’s initiation. Even though the studied population was small, it seems that if COVID-19 patients develop sinus bradycardia, a short lapse administration of either of these might be an effective treatment [18]. Theophylline relaxes smooth muscle and induces significant bronchodilation, provides a positive ionotropic effect, is a mild diuretic, and also shows cardiac and central nervous system (CNS) stimulant activities [10,19,20].

Theophylline´s relaxation of the bronchial smooth muscle and pulmonary blood vessels is mainly due to its activity as a phosphodiesterase inhibitor and adenosine receptor antagonist. Additionally, it has been proven to be an effective anti-inflammatory and immunomodulatory agent [12].

The outbreak of COVID-19 in late December 2019 has brought significant harm and challenges around the world [21]. SARS-CoV-2 has four structural proteins: spike glycoprotein (S), small envelope glycoprotein (E), membrane glycoprotein (M), and nucleocapsid protein (N) that are re-sponsible for viral replication [22,23]. SARS-CoV-2 binds mainly to the angiotensin-converting enzyme 2 (ACE2) receptor to enter the cells, in a way like SARS-CoV; however, this process is facilitated by the proteolytic cleavage of the S protein´s receptor binding domain (RBD) by the transmembrane protease serine 2 (TMPRSS2) [22,24,25]. The RBD on the S protein of SARS-CoV-2 exclusively recognizes the ACE2 receptor on the host cell [24,25]. After entering the cell, the positive-sense viral RNA genome is released to the cell cytoplasm and translated and replicated forming progeny genomes and subgenomic mRNAs. The latter is translated to membrane proteins, protein N, and a variety of accessory proteins [24]. The synthetized membrane proteins (S, M, and E) are then incorporated into the rough endoplasmic reticulum (RER) and transferred to the endoplasmic reticulum-Golgi intermediate compartment (ERGIC). The N proteins and the genomic RNA simultaneously form nucleocapsids, which fuse at the ERGIC. Finally, the viruses are transported by vesicle to the plasma membrane and released out of the cell via exocytosis [16,22,26,27]. In severe COVID-19 cases, SARS-CoV-2 affects the lower respiratory tract and infects type II pneumocytes, leading to apoptosis and loss of surfactant and causing fatal pneumonia [28,29]. It has been described that the development of acute respiratory distress syndrome (ARDS) increases the chances of death in elderly patients, especially if they suffer from metabolic syndrome, type 2 diabetes mellitus, or other serious chronic diseases [22,28]; ARDS severity is related to acute lung damage and systemic microcirculatory abnormalities [28]. On the other hand, SARS-CoV-2 infection causes the phenomenon coined as “cytokine storm”, which leads to the activation of macrophages and dendritic cells, as well as overproduction of cytokines in the center of COVID-19 classical general inflammation, increasing rapidly with pro-inflammatory tissue stress linked to systemic inflammation [28,30]. This issue is further described in subtopic 2.

Conceivably, the global spread of SARS-CoV-2 may continue for many months or years and will remain to have a serious impact on all human activities (health, social life, education, economy, etc.) [31,32]. The complexity and multifactorial characteristics of COVID-19 have encouraged different strategies to upgrade the clinical treatment of the disease. The use of social distancing, vaccines, antiviral drugs, and alternative clinical therapies has been paramount in preventing and treating COVID-19 in critically ill patients. Unfortunately, vaccinated people can still be infected by SARS-CoV-2, and too few antiviral treatments are available for COVID-19 patients. Therefore, the search for efficient treatments and beneficial dietary supplements continues to be an essential strategy in the fight against COVID-19 [33].

In this sense, we propose that the use of theophylline may provide health benefits against SARS-CoV-2, mainly through its bronchodilatory, immunomodulatory, anti-inflammatory, and antiviral effects proposed through in silico studies.

## 2. Immune Response in COVID-19 Patients

The immune response is induced by the presence of the virus detected by the host cell through pattern recognition receptors (PRR): Toll-like receptors (TLR), gene-I proteins inducible by retinoic acid (RIG-I), and NOD-like receptors (NLR) and other cytosolic viral sensors. When activated via adapter proteins, TLRs (primarily TLR3 and TLR7) activate the interferon regulatory factors (IRF3, IRF7) and a pro-inflammatory nuclear transcription factor (NF-κB). Consequently, the production and release of type I and III interferons (IFN-α/β and λ) are initiated, while NF-κB mediates the transcription of adhesion molecules, chemokines, colony-stimulating factors, and other cytokines that participate in the inflammatory response, including molecules that initiate neutrophil recruitment (Figure 1) [28,34,35,36,37,38].

Type I IFNs activate the JAK/STAT signaling pathway through IFNAR so that JAK1 and TYK2 kinase phosphorylate STAT 1 and 2. A complex among STAT 1/2 and IRF9 (the ISFG3 transcription factor) is formed and translocates to the nucleus to initiate transcription of IFN-stimulated genes. An effective type I IFN response in the early stages suppresses viral replication and its spread; during SARS-CoV and MERS-CoV infection, this response is suppressed, and its inactivity is associated with the severity of the disease [22,34,37,38]. Many SARS-CoV-2 proteins (nsp1, nsp3, nsp5, nsp6, nsp9, nsp13, nsp14, nsp15, orf3b, orf6, orf9b, N, and M) have been identified to inhibit this signaling pathway at different points, hindering a proper type I IFN response [22]. These proteins inhibit IRF3 by blocking its nuclear translocation and phosphorylation, consequently impeding IFNs synthesis in ribosomes and their transport to the membrane. They also diminish ISG nuclear translocation and ISGF3 complex formation and block several points of the IFN I-III/IFN-R (Tyk2, Jak1)/ISGF3 (STAT1, STAT2, IRF9)/ISG signaling pathways [22]. During SARS-CoV or MERS-CoV infection, a delay in the IFN type I inflammatory response, essential in early viral control, is observed, causing an exaggerated inflammatory response by neutrophils and macrophages. In SARS-CoV-2 infection, transmission can occur even through asymptomatic individuals, probably due to a delay in the innate immune response and corresponding to the main cause of fatality in severe acute respiratory syndrome [37]. It is also known that, in the cytosol, free viral RNA induces a conformational change in the RIG-I receptor, thus interacting with mitochondrial antiviral signaling proteins (AVMs) in the epithelium and with NLR receptors in myeloid cells to activate inflammasomes. In this sense, NLRP3 activation leads to the production of IL-1β and IL-18 dependent on caspase 1 [22,35]. In SARS-CoV-2 infection, the N protein and Orf3a both activate NLRP3, contributing to augmenting the inflammatory response (Figure 1) [22].

Alveolar macrophages respond promptly to a viral threat through phagocytosis of opsonized viral particles or apoptotic infected cells (efferocytosis) and/or through the production and release of inflammatory cytokines. Type I IFNs released through pathways described previously induce recruitment and differentiation of circulating precursor monocytes to alveolar macrophages and dendritic cells (Figure 1) [34,35,37].

In SARS-CoV-2, a phenomenon known as “cytokine storm” frequently develops and is characterized by a high production of pro-inflammatory cytokines that play an important role in the pathophysiology of the infection. Circulating cytokines found during the cytokine storm are IL-2, IL-4, IL-6, IL-7, IL-10, TNF-α, IFN-γ, IP-10, MCP-1, MIP-1A, granulocyte colony-stimulating factor (G-CSF), and granulocyte macrophage CSF (CG-CSF) [22,37,38,39]. Circulating levels of these cytokines are associated with greater morbidity during the infectious process. It has been established that IL-6 promotes the recruitment, differentiation, and activity of monocytes and T cells; in COVID-19 patients, an elevation in its circulating concentration was reported in 52% of the cases studied [37,40,41]. On the other hand, it has been described that TNF-α increases cytotoxic activity, leukocyte cytokine production, and endothelial cell activity (Figure 1) [35,40].

Neutrophils are among the first immune cells to respond to an infectious threat. They engulf virions and viral particles that are inactivated by proteolytic enzymes, antimicrobial peptides, and reactive oxygen species (ROS), and they also secrete granules with antimicrobial peptides. Interestingly, neutrophils release extracellular neutrophil traps (NETs) that immobilize pathogens and prevent their further spread. Although these mechanisms are helpful in mitigating infection, an excessive neutrophilic response can harm the host and lead to further lung damage (Figure 1) [34,36,38].

The innate immune response is the first defense mechanism activated. However, to complete viral clearance, stop viral replication, and effectively eliminate viral infection, the antiviral response and the adaptive immune response are necessary. The microenvironment created by the cytokine storm stimulates the differentiation and maturation of CD8^+^ cytotoxic T cells and CD4^+^ helper T cells [22,35,37,38,42]. In SARS-CoV and SARS-CoV-2 infection, the depletion of CD4^+^ T cells is associated with a reduction in the recruitment of lymphocytes, and of neutralizing and cytokine-producing antibodies, resulting in delayed viral clearance and strong immune-mediated interstitial pneumonitis [22,38]. CD4^+^ T cells also produce interleukins via NF-κB, especially IL-17, which recruits monocytes, neutrophils, the differentiation of Th17, and the production of cytokines and chemokines such as IL-1, IL-6, IL-8, IL-21, TNF-β, and MCP-1 [22,38]. Th17 participation in the pathogenesis of SARS-CoV-2 infection, especially in severe patients presenting a lower Treg/Th17 ratio, indicates that it negatively regulates Treg cells, promotes neutrophil migration and Th2 response, all these circumstances leading to an exaggerated immune response causing tissue damage and edema among the main complications of the disease (Figure 1) [22,43,44].

To limit infection in late stages and prevent reinfection in future times, the participation of the humoral immune response is required. In SARS-CoV infection, seroconversion is induced even as early as the fourth day of the infection onset, but in most patients, this response is evident by day 14. In preliminary studies in a patient with SARS-CoV-2, a specific IgM peak was observed on day 9 after the onset of the disease, changing to IgG predominance by the second week. A strong T-cell response correlated with higher neutralizing antibodies; however, a plasmatic Th2 cytokine pattern (IL-4, IL-5, IL-10) was observed in the group with higher lethality [22,37]. In severe cases of COVID-19, elevated levels of plasmatic CCR6+ Th17 were detected, indicating that the Th17 response is favored resulting in a pro-inflammatory response that might prompt pulmonary edema. Disproportionate elevation of the Th17 response is also observed in MERS-CoV and SARS-CoV infected patients [22,41,43,44]. Currently known memory T-cell responses against SARS-CoV are directed at structural proteins (such as the S, M, and N proteins). These responses last a long time, and the strongest reactions are directed against the spike protein [38,41]. Additional mechanisms activated during the response to SARS-CoV-2 and their potential as therapeutic targets are still unknown (Figure 1).

## 3. Immunomodulatory Effects of Theophylline

Theophylline is a well-known anti-inflammatory that exerts its effects through various mechanisms, with a notable history in the treatment of respiratory diseases with a strong inflammatory component, such as asthma and chronic obstructive pulmonary disease (COPD) and could therefore be beneficial in the treatment of COVID-19. In this sense, theophylline has been reported to have several interactions with different molecular targets (Figure 2 and Figure 3), that interact differently with several immunological routes once such targets are reached (Table 1).

In this context, several activities related to the immunological context of theophylline in humans and the importance in the treatment of COVID-19 must be mentioned.

### 3.1. Cytokine Inhibition

In COPD patients, continuous treatment with theophylline to maintain plasma levels of 9–11 mg/L for a 4-week period demonstrated a significant reduction in total inflammatory cells, predominantly in neutrophils, and lowering of IL-8, myeloperoxidase, and lactoferrin [46]. Additionally, theophylline produced a reduction in neutrophil chemotaxis induced by N-formyl-met-leu-phe and IL-8 [46]. This was also observed in airway smooth muscle cell cultures, where theophylline (10 µM) decreased TNF-α-induced IL-8 secretion via enhancing TNF-α-induced PP2A enzymatic activity [47]. Furthermore, the addition of low-dose theophylline (LDT) decreased IL-8 and IL-6 production in COPD patients with lung fibrosis [48] and enhanced the anti-inflammatory effects of standard steroids treatment by further decreasing IL-8 sputum levels in COPD patients [49]. In the long-term theophylline treatment in COPD patients (12 months), IL-8, TNF-α, and neutrophil sputum levels were progressively reduced [50]. In human peripheral blood mononuclear cells (PBMCs), theophylline treatment significantly reduced IL-1β and TNF-α production, but not IL-8 production induced by lipo-polysaccharide (LPS) or recombinant human IL-1β. The lack of effect on IL-8 production could be limited to this cell group since it is not consistent with other studies [51]. In atopic asthmatics, theophylline at 150 mg daily for 3 weeks reduced circulating serum levels of IL-4 and IL-5 [52]. In PBMCs from asthmatic children stimulated with house dust mite, theophylline at 20 µg/mL significantly reduced IL-5 and IL-3 production and lymphocyte proliferation [53]. Not only is theophylline able to inhibit IL-5 production, but it can also inhibit IL-5 induced degranulation in eosinophils, having a synergic effect when used in combination with procaterol (a β-adrenergic agonist) (Figure 1) [54].

An additional mechanism through which theophylline inhibits TNF-α-induced activation of NF-κB in a dose-dependent manner preventing the translocation of the transcription factor into the nucleus is by protecting the IκBα protein from degradation [55,56,57]. The resultant inhibition of the NF-κB pathway causes the suppression of pro-inflammatory cytokines, including TNF-α, IL-8, and GM-CSF [57]. The NF-κB and the p38 mitogen-activated protein kinases (MAPK) pathway are also regulated through the activity of dual-specificity phosphatases (DUSPs) [58]. Both DUSP1 and DUSP5 overexpression can inhibit p38 MAPK, and DUSP5 can also inhibit NF-κB [58,59,60]. Moreover, in COVID-19 patients, high levels of DUSP1 and DUSP5 have been detected, especially in severe cases [58]. It is possible that theophylline administration to COVID-19 patients increases the expression of DUSP1 and DUSP5 [58] to such an extent that they contribute to the treatment of the illness.

On the other hand, theophylline’s anti-inflammatory effects are also partially due to the increase in an anti-inflammatory cytokine (IL-10) secretion observed in asthmatics and COPD patients (Figure 1) [61,62].

### 3.2. Histone Deacetylase 2 Modulation

Another anti-inflammatory effect of theophylline is as a histone deacetylase (HDAC) activator. HDACs are a superfamily of enzymes that deacetylate histones to regulate gene expression [63,64]; specifically, HDAC2 modulates the inflammatory response in macrophages and monocytes by inhibition of NF-κβ, suppressing inflammatory genes and proteins (IL-8, GM-CSF) [63,64]. The effect of theophylline over HDAC2 is achieved at low doses (plasma concentration of 5 mg/L) [63,64,65]. In many respiratory diseases, including COPD, asthma, and viral infections such as by influenza A virus, the activity of HDACs is compromised [63,64,66]. Other components that decrease HDAC2 activity are reactive oxygen and nitrogen species (ROS and RNS), which are prominent in the inflammatory process of these diseases [67,68]. The mechanism by which theophylline regulates HDAC2 activity is by inhibiting oxidant-activated phosphoinositide-3-kinase-delta (PI3K-δ) [69]. In part, theophylline and other phosphodiesterase inhibitors (PDEIs) have been demonstrated to be HDAC2 modulators, which could be linked to IL-8, TNF-α, and ROS and RNS production since HDAC2 levels are linked to these [48,64,65,70]. As mentioned above, theophylline regulates IL-8 and TNF-α production and can also directly modulate ROS production in human monocytes via PDEI [71]. Through computational methods, cellular targets for SARS-CoV-2 miRNAs were identified. Among the targets, HDAC2 was identified to be the target for SARS-CoV-2-mir-D10-5p and SARS-CoV-2-mir-D6-3p. As an HDAC2 activator, theophylline might be influencing this pathway (Figure 1) [72].

### 3.3. Matrix Metalloproteinases Suppression

Matrix metalloproteinases (MMPs) are a family of Zn^2+^ and Ca^2+^ dependent proteolytic enzymes that participate in tissue remodeling through the ability to degrade extracellular and membrane basement components. They are secreted by numerous cells (inflammatory cells, epithelial cells, stromal cells, fibroblasts, etc.) [73,74,75]. The dysregulation of MMPs can be linked to a variety of diseases, and therefore, the identification of elements that could alter MMP activity is of great interest. Many inflammatory components participate in the regulation of MMP activity, including inflammatory cytokines (IL-1β, IL-6, TNF-α, and IFN-γ), ROS, and RNS [73,76]. MMPs’ participation in many respiratory diseases has been described, including tissue remodeling and acute lung injury [73,74,75]. Recently, an association between higher plasmatic MMP-2 and MMP-9 levels and COVID-19 severity was found, and it has even been proposed as a potentially accurate prognosis factor [77]. In human lung fibroblast, theophylline treatment can inhibit RNS-induced MMP-2 and MMP-9 release via the NF-κβ/TGF-b1 pathway and probably through HDCA2 activity as well [65]. Theophylline suppression of MMPs could be an additional therapeutic mechanism that might benefit COVID-19 patients (Figure 1).

### 3.4. SIRT1 Activation

The overactivation of PARP1 quickly causes NAD^+^ depletion [78,79,80]. NAD^+^ is involved in multiple metabolic processes as a cofactor, catalyzing electron transfer in metabolic reduction-oxygenation reactions, such as ATP production. Its lack or significant reduction could cause an energy crisis that would lead to cell apoptosis [78,79,80]. NAD^+^ depletion could also lead to decreased sirtuin 1 (SIRT1) activity [78,79,80]. SIRT1 is an NAD^+^-dependent deacetylase of nuclear proteins that regulates gene expression cytokines, tumor suppressors, and proto-oncogenes and modulates inflammatory processes, cell survival, and apoptosis [78,79,80]. COVID-19 pathophysiology includes not only increased ROS production that leads to depletion of NAD^+^ but also decreased expression of SIRT1 [78,79,80,81]. PARP-1 activation and SIRT1 downregulation are important players in the inflammatory processes of many pulmonary diseases such as COPD, asthma, and some viral infections and have now been described in SARS-CoV-2 [78,79,80,81,82]. In human pulmonary epithelial cells, theophylline treatment inhibited PARP-1, preventing NAD^+^ depletion [78], and in COPD patients, treatment with prednisolone together with theophylline increased SIRT1 expression [82,83]. Seemingly, targeting PARP-1 and SIRT1 with theophylline could also be beneficial in the treatment of COVID-19 patients (Figure 1).

### 3.5. mTOR Signaling Inhibition

Another prominent drug target gaining interest in the pathophysiology of COVID-19 is the mechanistic target of rapamycin (mTOR) pathway. mTOR is a serine/threonine protein kinase involved in the regulation of numerous intracellular processes, for instance, cell metabolism, proliferation, growth, and survival [84,85,86]. In order to exert its activity, mTOR binds to the multiprotein complexes mTOR complex 1 and complex 2 (mTORC1 and mTORC2); each complex participates in distinct signaling functions [84,85,86]. mTOR pathway has been characterized in immune cells. It is known that activation of mTORC1 controls IL-15-activated NK cell cytotoxicity and controls BCL6 expression to control B cells in the germinal line [84]. Inhibiting mTORC1 enhances dendritic cells’ T-cell stimulatory activity and autophagy of macrophages and reduces antigen-specific memory B cell population after B cell activation [84]. In some severe cases of COVID-19, it was speculated that prior to exposure to other coronaviruses and because of antigenic epitope heterogeneity, antibody-dependent enhancement (ADE) could have developed [84,87]. Enhancement has been previously described in SARS-CoV, caused by the development of anti-spike protein antibodies, causing the infection of immune cells [84,87,88,89]. In the early stages of COVID-19 infection, blocking the activation of memory B cells through mTOR inhibitors could reduce cross-reactive antibodies to SARS-CoV-2 and mitigate the more severe symptomology [84]. On the other hand, mTORC2 regulates neutrophil and mast cells’ chemotaxis and cell polarity [84]. Furthermore, the activation of the inflammasome NLRP3 and macrophage pyrosis is regulated through both mTORC1 and mTORC2 [90]. The mTOR pathway has been shown to be modulated through the increment in intracellular levels of cAMP [85,91,92]. The inhibition of mTORC1 seems to be mediated by cAMP via a PKA or PKB-dependent mechanism since cAMP activates the negative regulator of mTORC1, TSC1/2 [91,92]. This mechanism is not solely responsible for mTORC1 inhibition but can also be regulated by cAMP disturbing mTOR and Rheb co-localization through a Rag GTPase-dependent mechanism without the involvement of TSC1/2 [92]. Additionally, prolonged elevation of cAMP leads to mTORC2 inhibition and decreased activity of mTOR (Figure 1) [92].

Another concern of the mTOR pathway in the pathophysiology of SARS-CoV-2 is its participation in the viral life cycle. Viruses hijack the host cells transcription and translation machinery during its life cycle and mTOR is known to contribute to both DNA and RNA virus replication by acting on phosphoinositide 3-kinase (PI3K), Akt, or even mTOR directly [86,93]. Theophylline, by producing cAMP increments, activates PKA and PKB, inhibiting mTOR [91]. The inflammatory attenuation produced by mTOR signaling inhibition, as well as the interruption of the viral replication, are promising mechanisms of action of theophylline that might contribute to the treatment of SARS-CoV-2 pathophysiology (Figure 1).

## 4. In Silico Screening of Theophylline: Therapeutic Targets and Potential Agents

Recently published works have suggested theophylline´s potential antiviral activity using in silico molecular dynamics and molecular docking, and therefore originally performed studies are included herein. In silico studies to predict whether a ligand (drugs, biomolecules, or plant-derived compounds) can produce the expected biological effect are mainly done by structure-based drug design methods. Among the most used methods, we find the molecular docking and molecular dynamics simulation techniques [94,95]. Such techniques are widely used due to the range of applications in the analysis of molecular recognition events, such as binding energy, molecular interactions, and induced conformational changes [94]. To determine if theophylline is a feasible candidate to be used in the treatment against COVID-19, it was analyzed using structure-based techniques. Studies exploring this subject have analyzed the interactions of theophylline with different protein targets, for instance, the chymotrypsin-like protease protein (3CLpro) and the nucleocapsid protein (N) [15,96]. The interaction between the spike protein and ACE2 has gained particular interest for its potential to be used as a pharmacological target and has inspired the exploration of caffeine, another methylxanthine, as a possible pharmacological tool [13,97]. To expand the knowledge in this regard, we performed in silico experiments exploring the interaction of theophylline´s affinity for spike and envelope proteins. Original results from our team are shown in section theophylline affinity with spike and envelope proteins.

### 4.1. 3CLpro and Theophylline

The 3CLpro protein is a cysteine protease comprised of three domains and generates functional polypeptides playing an important role in viral transcription and replication. The active sites of 3CLpro are S1ʹ, S1, S2, and S4 and are highly conserved in different coronaviruses such as MERS-CoV, SARS-CoV, and SARS-CoV-2 [98]. For the catalytic function of 3CLpro, the following amino acids are essential: Cys145, Glu166, and His41 [14]. Furthermore, there is no closely related homolog of 3CLpro in humans, making it an attractive target for innovative anti-COVID-19 drugs (Figure 1) [98].

In 2022, through molecular docking and molecular dynamics simulations, Elzupir et al. studied the interaction of bromotheophylline, pentoxifylline, and theophylline with the 3CLpro protein of the SARS-CoV-2 virus. These drugs formed a hydrogen bond with key residues for the inactivation of the 3CLpro protein; bromotheophylline and theophylline bound to amino acid Glu166, while pentoxifylline bound to Cys145. Theophylline forms a hydrogen bond with Glu166 and also establishes van der Waals forces with the amino acids Glu166, Cys145, and His41 [15]. Additionally, by means of molecular dynamics methods, the stability of the complexes between the 3CLpro protein and bromotheophylline, pentoxifylline, and theophylline was verified [15]. This study shows that theophylline and its derivatives are possible inhibitors of the enzymatic activity of the 3CLpro protein (Figure 1).

### 4.2. N-Protein and Theophylline

The N-protein conforms to the RNA package and participates in the release of viral particles. It contains two RNA-binding domains: the N-terminal domain (NTD) and the C-terminal domain (CTD), linked by a serine/arginine-rich domain (SRD) [99]. Due to the positively charged amino acids, the CTD and NTD domains bind to the viral RNA genome, and, in addition, the SRD improves the oligomerization capacity [99]. It has been shown that the N-protein promotes the activation of the NLRP3 inflammasome to induce hyperinflammation, facilitating the maturation of pro-inflammatory cytokines and aggravating lung lesions [100]. Several different inhibitors of this protein have been studied, and the interaction between N-protein and theophylline has been analyzed. In this context, Sarma et al. evaluated different bronchodilators able to inhibit the binding of viral RNA to the NTD of N-protein using in silico techniques and established that theophylline stably binds to the site that inhibits the interaction between RNA and protein N (Docking score: −3.763 kcal/mol and MM-GBSA −39.464 kcal/mol) [96]. Furthermore, they found that a derivative of theophylline (ZINC3118440) is a plausible inhibitor of the NTD site of the N-protein (Figure 1) [96].

### 4.3. Theophylline Affinity with Proteins Spike and Envelope

Our team performed molecular docking to predict the binding affinity between theophylline and two pharmacological targets, the S and E proteins.

Docking directed to sites of interest was carried out. In the case of the spike protein, the RBD binding site was explored, and for protein E, the affinity and mode of binding to this ion channel were predicted. The following (Figure 4) shows the binding modes of theophylline with the spike protein (Figure 4A) and protein E (Figure 4B). The delta G between the spike protein and theophylline was −5.4 kcal/mol, and Figure 4A illustrates that the binding is formed by four hydrogen bonds with amino acids Arg493B, Phe377A, Thr376A, and Phe375A, plus a Pi–Pi stacked interaction with Phe377A and Van de Waals forces with Phe456B, Tyr489B, Phe490B, and Ala484B. The structure and conformation of the spike protein that has been used to carry out the binding prediction belongs to the 7WK3 crystal that corresponds to the open conformation of the Omicron variant. According to molecular docking, theophylline is capable of binding to the spike protein of the Omicron variant.

In this sense, we may conclude that the spike protein and protein E are of pharmacological interest since they are the main structural proteins of the SARS-CoV-2 virus that participate in mechanisms of the viral cycle and are also involved in the inflammatory processes [95,104]. Interestingly, theophylline binds to amino acid Ala484, a mutation (E484A) found in the Omicron variant responsible for the greater resistance to neutralization by Bamlanivimab shown by this variety [105]. Theophylline also binds to amino acids close to Val25, responsible for inhibiting the ion channel function and oligomerization (Figure 1) [106,107]. Recently, in silico studies conducted by Rolta et al. showed an interesting binding affinity between caffeine, methylxanthine, theobromine, and theophylline with SARS-CoV-2 spike protein and S1 receptor-binding domain (S1RBD). Theoretically, all the above-mentioned phytocompounds could potentially prevent the binding of SARS-CoV-2 to the ACE2 receptor, sensibly diminishing the virus’s infectious capacity. Notwithstanding these promising results, these methylxanthine derivatives must be probed in vitro and in vivo to confirm their therapeutic benefits against SARS-CoV-2 [108].

## 5. Quantitative-Systems Pharmacology Analysis

Theophylline plasma concentrations of 10–20 µg/mL are therapeutically relevant, with maximal bronchodilatory effects observed at 10 µg/mL [10,109,110]. Nevertheless, the anti-inflammatory effects of theophylline develop at lower doses, around 5–10 µg/mL [10,109,110]. To avoid unwanted side effects the range can be narrowed to 5–15 µg/mL, even though toxicity might be observed at concentrations as low as 15 µg/mL [10,109,110].

In this context, using PK-Sim (v 8.0) [111], we performed a quantitative-systems pharmacology (QSP) analysis where we simulated two scenarios: the first one was a healthy patient (>70 years old) that received two different doses of theophylline intravenously (IV), and the second was a patient with severe renal impairment (glomerular filtration rate [GFR] = 5 mL/min/100 g organ) [112] that also received theophylline (Figure 5 and Figure 6).

Massive albuminuria and subsequent development of proteinuria seen in SARS-CoV-2 show that renal involvement is common in this illness; it has been established that renal damage ranges from mild proteinuria to advanced acute kidney injury. SARS-CoV-2 binds to ACE2 receptors in many organs of the body, especially the proximal tubules of the kidney that have more ACE2 receptors than the lungs. Therefore, the virus has a great impact on the renal tubules [113], thus hindering theophylline clearance, leading to intoxication and aggravating existing renal failure that may lead to death. Notwithstanding, theophylline reaches therapeutic levels in peripheral blood during the first hours of administration, and the lung is one of the first organs reached by this methylxanthine. In this sense, theophylline seems to be at the interstisium during the first hours after administration, probably indicating that, if renal failure were present, it might be overcome before theophylline’s toxic effects become risky.

## 6. Conclusions

Theophylline seems to be a relevant candidate for the treatment of COVID-19 patients since it aims several immunological targets involved in the pathophysiology of the disease. Through various mechanisms theophylline down-regulates the overactivation of the inflammatory processes activated by SARS-CoV-2. Additionally, it might have antiviral potential that could hinder the pathogenesis of the virus. Considering that many viral respiratory infections mount a similar host immune response [34,35], as was the case with the influenza, SARS-CoV, MERS or SARS-CoV-2 viruses; therefore, theophylline should be under consideration as a prime candidate in the drug discovery process for any future viral pathogens. Furthermore, COVID-19 is now also known to have long lasting clinical presentation, such as with “long-haul” cases [25,114,115,116], extending past the acute infection phase, and theophylline has proven to be effective in the chronic management of respiratory diseases, particularly those with an inflammatory component, such as asthma and COPD, and could also be repurposed for this case. Theophylline, offering multiple advantages with a known clinical use and an affordable price, should not be overlooked in the search for potential treatments for COVID-19 patients.

## Figures and Tables

**Figure 1 ijms-23-04167-f001:**
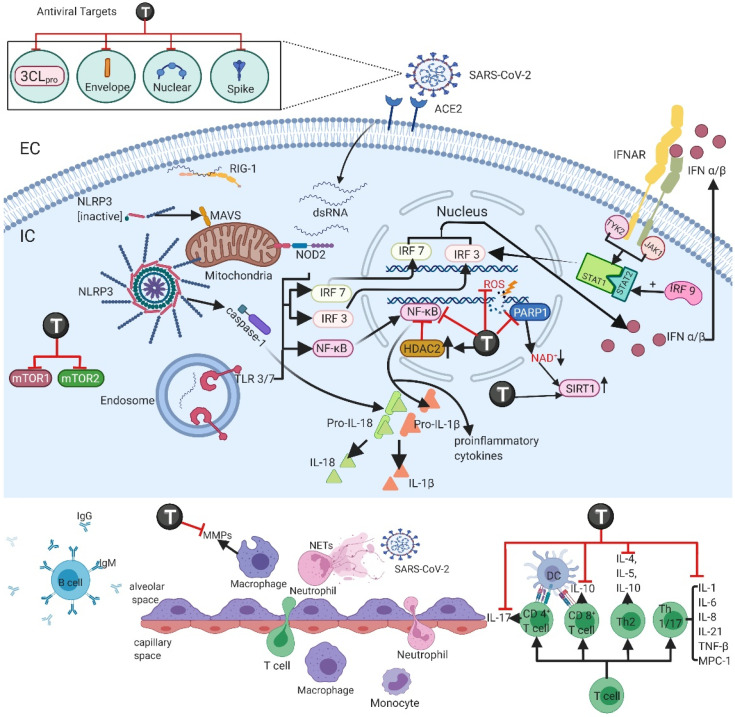
SARS-CoV-2 enters the cell through ACE2 found in the membrane of type II pneumocytes. The immune response is initiated through the activation of PRRS (TLR, NLR, RIG-1) located in the membrane and cytosol interacting with the viral RNA. The stimulation of these receptors leads to the activation of transcription regulating factors such as IRF 3/7 and NF-kB, which leads to the production of interferons type I/III and other pro-inflammatory cytokines. Theophylline modulation of HDAC2 inhibits NF-kB and inflammatory genes. DNA damage produced by ROS production activates PARP1, which leads to NAD^+^ depletion, decreasing SIRT1 activity. Theophylline inhibits PARP1, decreases ROS production, and enhances SIRT1 expression and activity. mTOR inhibition by theophylline contributes to attenuating the inflammatory response. The production of these cytokines initiates the recruitment of neutrophils and recruitment and differentiation of monocytes into alveolar macrophages. The cytokine storm microenvironment produced as a result of the viral infection stimulates the activation and differentiation of cytotoxic CD8^+^ and helper CD4^+^ T cells necessary for viral clearance, initiating the production of cytokines, recruitment of monocytes and neutrophils, promoting the immune response. The proposed antiviral effects of theophylline are highlighted. In order to successfully limit the viral infection in later phases of the disease and to prevent future reinfections, the participation of the humoral response and production of antibodies is needed. Abbreviations: T—theophylline; PRRs—pattern recognition receptors; TLR—Toll-like receptors; RIG-1—gene-I proteins inducible by retinoic acid; NLR—NOD-like receptors; IRF 3/7—interferon regulatory factors; NF-κB—pro-inflammatory nuclear transcription factor; HDAC2—histone deacetylase 2; ROS—reactive oxygen species; PARP1—poly ADP ribose polymerase 1; SIRT1—sirtuin 1; mTOR—mechanistic target of rapamycin pathway; MMPs—matrix metalloproteinases.

**Figure 2 ijms-23-04167-f002:**
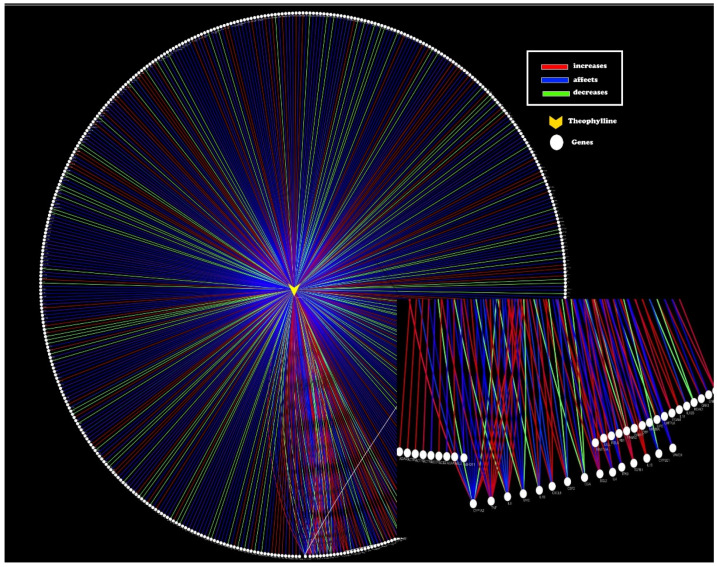
Pharmacological network biology analysis of theophylline-genes interaction. Node in yellow represents theophylline and white nodes represent genes that are modified to different degrees: somehow affected (blue), increased (red) or decreased activity (green) according to the Comparative Toxicogenomics Database [45].

**Figure 3 ijms-23-04167-f003:**
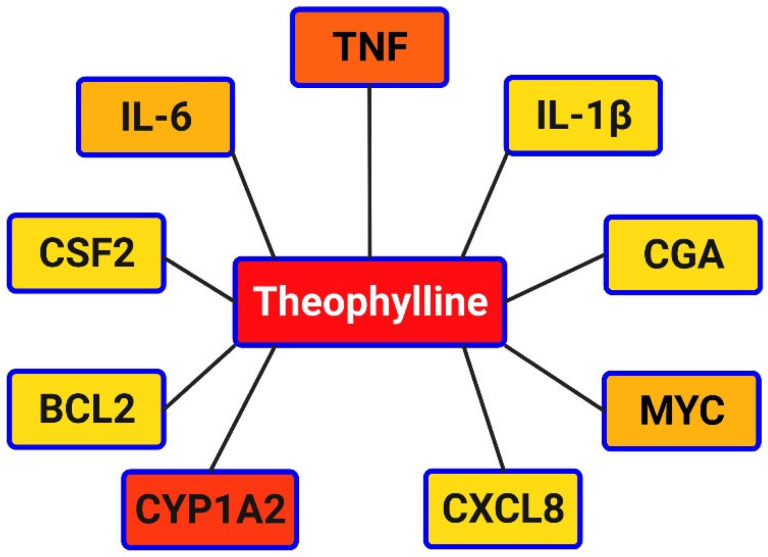
Network result from the most connected nodes. Color shades represent connection degrees from most connected (yellow) to less connected (red).

**Figure 4 ijms-23-04167-f004:**
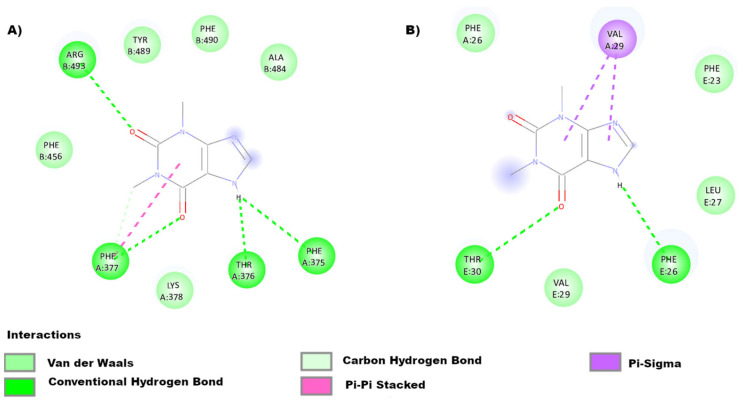
Molecular docking between theophylline and SARS-CoV-2 proteins (spike and protein E). In both couplings, we observe Van der Waals forces. (**A**) The interactions between theophylline and the spike protein show 4 hydrogen bonds and a Pi–Pi stacked interaction. (**B**) The relation with the E protein has Pi–sigma interactions and two hydrogen bonds. Molecular docking of protein E with theophylline showed that the delta G was −5.0 kcal/mol. (**B**) illustrates that the xanthine binds to amino acids Phe23E, Phe26A, Phe26E, Leu27E, Val29A, Val29E, and Thr30E (methodology in Appendix A [101,102,103]).

**Figure 5 ijms-23-04167-f005:**
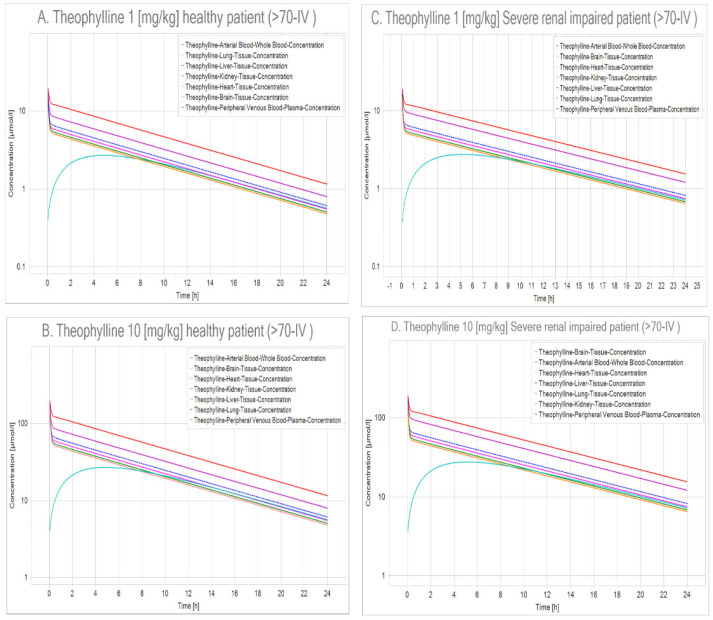
Physiologically based pharmacokinetic model (PBPK) simulation of theophylline at two concentrations modeled in a healthy patient and a patient with renal impairment. (**A**) Healthy patient dosed with 1 mg/kg, (**B**) Healthy patient that received 10 mg/kg, (**C**) Renal impaired patient administered 1 mg/kg. (**D**) Renal impaired patient dosed with 10 mg/kg. When results of the healthy patient were compared with those of the renal insufficiency patient, a slight increase in the time of theophylline body retention was observed in all organs studied (clearance reduction). Tissue concentration curves for the lungs (dotted blue curves) illustrate that the longest time for theophylline removal takes place in this tissue (Appendix A Appendix A).

**Figure 6 ijms-23-04167-f006:**
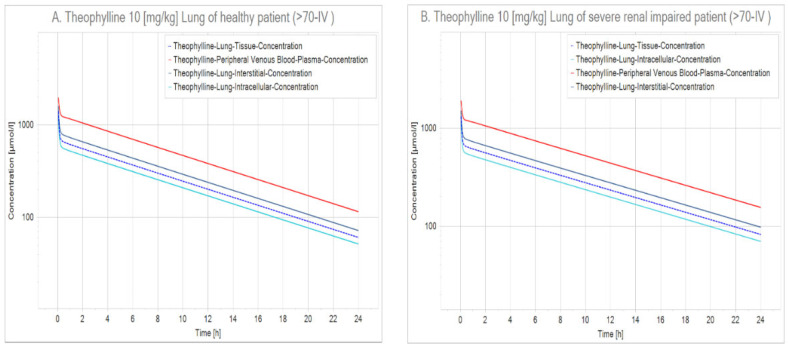
Physiologically based pharmacokinetics model (PBPK) simulation of 10 mg/kg theophylline in the lungs. (**A**) Kinetics in the lung of a healthy patient. (**B**) Kinetics in the lung of a patient with renal impairment. This figure shows that theophylline clearance is sensibly delayed in patients with impaired renal function. When comparing both charts, a decrease in the inclination of the lung concentration curves can be observed for the patient with renal failure. Noteworthy, renal elimination is the main route to excrete theophylline and consequently, a reduced clearance increases the time the drug remains in the body increasing the possibility of unwanted side effects (Appendix A Appendix A).

**Table 1 ijms-23-04167-t001:** Enrichment analysis of pathways altered by theophylline in humans.

Pathway	Pathway ID	Annotated Genes Quantity
Interleukin-10 signaling	REACT:R-HSA-6783783	10
Immune system	REACT:R-HSA-168256	45
Metabolism of proteins	REACT:R-HSA-392499	38
Interleukin-4 and 13 signaling	REACT:R-HSA-6785807	12
IL-17 signaling pathway	KEGG:hsa04657	11
NOD-like receptor signaling pathway	KEGG:hsa04621	13
Cytosolic DNA-sensing pathway	KEGG:hsa04623	9
Post-translational protein modification	REACT:R-HSA-597592	28
Inflammatory bowel disease (IBD)	KEGG:hsa05321	9
Jak-STAT signaling pathway	KEGG:hsa04630	12
Jak-STAT signaling pathway	KEGG:hsa04630	8
AGE-RAGE signaling pathway in diabetic complications	KEGG:hsa04933	7
Signaling by interleukins	REACT:R-HSA-449147	12
Longevity regulating pathway-multiple species	KEGG:hsa04213	6
Transcriptional misregulation in cancer	KEGG:hsa05202	8
Diseases of signal transduction	REACT:R-HSA-5663202	10
Cytokine signaling in immune system	REACT:R-HSA-1280215	27
Signaling by interleukins	REACT:R-HSA-449147	24
Innate immune system	REACT:R-HSA-168249	32
Interleukin-4 and 13 signaling	REACT:R-HSA-6785807	15
Cytokine-cytokine receptor interaction	KEGG:hsa04060	15
IL-17 signaling pathway	KEGG:hsa04657	11
Hematopoietic cell lineage	KEGG:hsa04640	11
Signal transduction	REACT:R-HSA-162582	33

## Data Availability

Not applicable.

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
