# Peer review of "Theophylline: Old Drug in a New Light, Application in COVID-19 through Computational Studies"

_ijms, 2022, doi:10.3390/ijms23084167_

Round 1

Reviewer 1 Report

Theophylline is 100 years old drug and its use is in decline due to complex pharmacokinetics, side effects and drug interactions, despite it is cheap and still in use in many countries. 

The review is very well written concerning COVID-19. However, information in regard to theophylline is incommensurate, as the title focus on theophylline. 

The abstract is not informative enough. It is written too generally. 

Very scarce information is presented about complicated pharmacokinetics of theophylline and, accordingly, its restricted indications and variety of drug interactions. 

There is no information about clinical use of theophylline in the treatment of COVID-19.  The authors cited two papers No 14 and 16, however, discussion is missing. 

The authors also describe their own results in "4.3. Theophylline affinity with proteins Spike and Envelope", however only results are presented without any method described. 

The authors should discuss and cite the already published paper of Rolta et al. "Methylxanthines as Potential Inhibitor of SARS‑CoV‑2: an In Silico Approach." Current Pharmacology Reports 2022 Mar 8: 1-22. 

The article No13 of Amin O. Elzupir is in fact published 2022, but not in 2020, as is stated in the presented paper. 

All the cited literature must be revised as there are mistakes in 43 of 108 cited papers. 

Author Response

AUTHORS´ANSWERS TO REVIEWERS

MAJOR REVISION. 

REVIEWER 1

Theophylline is 100 years old drug and its use is in decline due to complex pharmacokinetics, side effects and drug interactions, despite it is cheap and still in use in many countries.  

 The review is very well written concerning COVID-19. However, information in regard to theophylline is incommensurate, as the title focus on theophylline.  

Q= The abstract is not informative enough. It is written too generally. 

Answer: Thank you for your observation!

In accordance with it, we included the following sentences as specified bellow.

In the ABSTRACT we added (page 1, line 34-38) the following:

Nevertheless, it is essential to understand how this compound could behave against such a disease, not only at a pharmacodynamic but also a pharmacokinetic level. In this sense, the quickest approach in drug discovery is through different computational methods, either from network pharmacology or from quantitative systems pharmacology approaches.

In the same section, lines 39-43 now read:

“…the treatment of COVID-19 patients, since it seems to be a relevant candidate by aiming several immunological targets involved in the pathophysiology of the disease. Theophylline downregulates the inflammatory processes activated by SARS-CoV-2 through various mechanisms, and herein, they are discussed by reviewing computational simulation studies and its different applications and effects.”

We included another KEY WORD (page 1, line 45) typed in red:

quantitative systems pharmacology

Q= Very scarce information is presented about complicated pharmacokinetics of theophylline and, accordingly, its restricted indications and variety of drug interactions. 

A= Thank you for your comments, we added some sentences to the manuscript to expand this topic.

In the INTRODUCTION section (page 2, line 55-68), we added:

Special consideration in the administration of theophylline must be taken in circumstances that can represent reduced drug clearance, as for instance patients with liver or pulmonary disease (pneumonia, COPD) and heart failure. In contrast, an increased clearance can be expected in children <16 years and smokers [10]. Besides, several drug interactions should be considered when theophylline is administrated, for example decreased clearance in association with erythromycin, quinolones, allopurinol, cimetidine, serotonin uptake inhibitors and the 5-lipoxygenase inhibitor zileuton, and a higher metabolism when administered in conjunction with phenytoin, phenobarbital and rifampicin [10]. On the other hand, common side effects observed with theophylline treatment are headaches, gastrointestinal discomfort, insomnia, nausea, vomiting and at higher doses serious side effects such as seizures and cardiac arrythmias, primarily due to antagonism of adenosine 1 receptor (A1Rs), might develop. Low dose, slow-release treatment seems to be well tolerated in long term treatments for COPD and asthmatic patients, reducing exacerbations as well as the probability of serious side effects [10, 11].

  1. Barnes, P. J. Theophylline. Am J Respir Crit Care Med. 2013, 188(8), pp. 901-906. doi: 10.1164/rccm.201302-0388pp.
  2. Zhou, Y.; Wang, X.; Zeng, X.; Qiu, R.; Xie, J.; Liu, S.; Zheng, J.; Zhong, N.; Ran, P. Positive benefits of theophylline in a randomized, double-blind, parallel-group, placebo-controlled study of low-dose, slow-release theophylline in the treatment of COPD for 1 year. Respirology. 2006, 11(5):603-10. doi: 10.1111/j.1440-1843.2006.00897.x.

Q= Very scarce information is presented about complicated pharmacokinetics of theophylline and, accordingly, its restricted indications and variety of drug interactions. 

A= Because of the reviewer´s concern, we decided to include in this amended manuscript a quantitative-systems pharmacology analysis that was not incorporated before after a discussion with the editor. However, we believe that the section merits the inclusion in the Review as it would undeniably enrich the work. Below, we further discuss the reasoning behind this consideration.

In the present review we focus only on the potential use of theophylline for COVID-19. Therefore, as part of our proposal, we performed an in silico simulation, where we used pharmacokinetic information (already available) for this xanthine and simulated the consequences of its I.V. administration to a 70-year-old patient without renal damage- and a 70-year-old patient with renal failure. Such an approach provided 2 important data: on one side, it was shown in silico, that theophylline may have good bioavailability in the lung and, on the other side, that I.V. may be the best route for vulnerable patients. Data for simulation were obtained from the following 4 references and the simulation was performed with Pk-Sim software – 3:

  1. Britz, H.; Hanke, N.; Volz, A. K.; Spigset, O.; Schwab, M.; Eissing, T.; Wendl, T.; Frechen, S.; Lehr, T. Physiologically-Based Pharmacokinetic Models for CYP1A2 Drug-Drug Interaction Prediction: A Modeling Network of Fluvoxamine, Theophylline, Caffeine, Rifampicin, and Midazolam. CPT Pharmacometrics Syst Pharmacol. 2019, 8(5),296-307. doi: 10.1002/psp4.12397.
  2. Schlender, J. F.; Meyer, M.; Thelen, K.; Krauss, M.; Willmann, S.; Eissing, T.; Jaehde, U. Development of a Whole-Body Physiologically Based Pharmacokinetic Approach to Assess the Pharmacokinetics of Drugs in Elderly Individuals. Clin Pharmacokinet. 2016, 55(12),1573-1589. doi: 10.1007/s40262-016-0422-3.
  3. Hanke, N.; Türk, D.; Selzer, D.; Ishiguro, N.; Ebner, T.; Wiebe, S.; Müller, F.; Stopfer, P.; Nock, V.; Lehr, T. A Comprehensive Whole-Body Physiologically Based Pharmacokinetic Drug-Drug-Gene Interaction Model of Metformin and Cimetidine in Healthy Adults and Renally Impaired Individuals. Clin Pharmacokinet. 2020, 59(11), pp. 1419–1431. https://doi.org/10.1007/s40262-020-00896-w
  4. Willmann, S.; Lippert, J.; Sevestre, M.; Solodenko, J.; Fois, F.; Schmitt, W. PK-Sim®: a physiologically based pharmacokinetic ‘whole-body’ model. BIOSILICO. 2003, 1(4), pp. 121-124. doi: DOI: 10.1016/S1478-5382(03)02342-4.

With this consideration in mind, we added the section 5. Quantitative-systems pharmacology analysis (page 12-14, line 514-553).

  1. Quantitative-systems pharmacology analysis

Theophylline plasma concentrations of 10-20 µg/mL are therapeutically relevant, with maximal bronchodilatory effects observed at 10 µg/mL [10, 106, 107]. Nevertheless, the anti-inflammatory effects of theophylline develop at lower doses, around 5-10 µg/mL [10, 106, 107]. To avoid unwanted side effects the range can be narrowed to 5-15 µg/mL, even though toxicity might be observed at concentrations as low as 15 µg/mL [10, 106, 107].

In this context, using PK-Sim (v 8.0) [108] we performed a quantitative-systems pharmacology (QSP) analysis where we simulated two scenarios: the first one was a healthy patient (>70 years old) that received two different doses of theophylline intra-venously (IV) and the second was a patient with severe renal impairment (glomerular filtration rate [GFR] = 5 ml/min/100 g organ) [109] that also received theophylline (Figures 5 and 6).

Figure 5. Physiologically based pharmacokinetic model (PBPK) simulation of theophylline at two concentrations modeled in a healthy patient and a patient with renal impairment. (A) Healthy patient dosed with 1 mg/kg, (B) Healthy patient that received 10 mg/kg, (C) Renal impaired patient administered 1 mg/kg. (D) Renal impaired patient dosed with 10 mg/kg. When results of the healthy patient were compared with those of the renal insufficiency patient, a slight increase in the time of theophylline body retention was observed in all organs studied (clearance reduction). Tissue concentration curves for the lungs (dotted blue curves) illustrate that the longest time for theophylline removal takes place in this tissue.

Figure 6. Physiologically based pharmacokinetics model (PBPK) simulation of 10 mg/kg theophylline in the lungs. (A) Kinetics in the lung of a healthy patient. (B) Ki-netics in the lung of a patient with renal impairment. This figure shows that theophyl-line clearance is sensibly delayed in patients with impaired renal function. When comparing both charts, a decrease in the inclination of the lung concentration curves can be observed for the patient with renal failure. Noteworthy, renal elimination is the main route to excrete theophylline and consequently, a reduced clearance will increase the time the drug remains in the body increasing the possibility of unwanted side effects.

Massive albuminuria and subsequent development of proteinuria seen in SARS-CoV-2 show that renal involvement is common in this illness; it has been established that renal damage ranges from mild proteinuria to advanced acute kidney injury. SARS-CoV-2 binds to ACE2 receptors in many organs of the body, especially the proximal tubules of the kidney that have more ACE2 receptors than the lungs. There-fore, the virus has a great impact on the renal tubules [110], thus hindering theophylline clearance, leading to intoxication and aggravating existing renal failure that may lead to death. Notwithstanding, theophylline reaches therapeutic levels in peripheral blood during the first hours of administration, and the lung is one of the first organs reached by this methylxanthine. In this sense, theophylline seems to be at the interstisium during the first hours after administration, probably indicating that, if renal failure were present, it might be overcome before theophylline´s toxic effects become risky.

  1. Barnes, P. J. Theophylline. Am J Respir Crit Care Med. 2013, 188(8), pp. 901-906. doi: 10.1164/rccm.201302-0388pp.
  2. Vassallo, R.; Lipsky, J. J. Theophylline: Recent Advances in the Understanding of Its Mode of Action and Uses in Clinical Practice. Mayo Clin Proc. 1998, 73(4), pp. 346-354. doi: 10.1016/s0025-6196(11)63701-4.
  3. Senchina, D. S.; Hallam, J. E.; Kohut, M. L.; Nguyen, N. A.; Perera, M. A. Alkaloids and athlete immune function: caf-feine, theophylline, gingerol, ephedrine, and their congeners. Exerc Immunol Rev. 2014, 20, pp. 68–93.
  4. Willmann, S.; Lippert, J.; Sevestre, M.; Solodenko, J.; Fois, F.; Schmitt, W. PK-Sim®: a physiologically based pharmaco-kinetic ‘whole-body’ model. BIOSILICO. 2003, 1(4), pp. 121-124. doi: DOI: 10.1016/S1478-5382(03)02342-4.
  5. Hanke, N.; Türk, D.; Selzer, D.; Ishiguro, N.; Ebner, T.; Wiebe, S.; Müller, F.; Stopfer, P.; Nock, V.; Lehr, T. A Compre-hensive Whole-Body Physiologically Based Pharmacokinetic Drug-Drug-Gene Interaction Model of Metformin and Cimetidine in Healthy Adults and Renally Impaired Individuals. Clin Pharmacokinet. 2020, 59(11), pp. 1419–1431. https://doi.org/10.1007/s40262-020-00896-w
  6. Ahmadian, E.; Hosseiniyan Khatibi, S. M.; Razi Soofiyani, S.; Abediazar, S.; Shoja, M. M.; Ardalan, M.; Zununi Vahed, S. Covid-19 and kidney injury: Pathophysiology and molecular mechanisms. Rev Med Virol. 2021, 31(3). https://doi.org/10.1002/RMV.2176

Even though we believe the inclusion of (section 5). Quantitative systems analysis would enrich our work, we have independently addressed the concerns of the pharmacokinetics of theophylline.

Q= There is no information about clinical use of theophylline in the treatment of COVID-19.  The authors cited two papers No 14 and 16, however, discussion is missing.  

A= Thank you for your observation, we appreciate your fine comment! The information concerning these two references was expanded and the following text was added to the manuscript:

In the INTRODUCTION we added (page 2, line 73-93):

Furthermore, recent studies have contributed to establish theophylline´s great therapeutic potential in the COVID-19 treatment. Wall et al., carried out a retrospective study in COVID-19 patients that required oxygen and received either theophylline or pentoxifylline. Patients with a history of asthma or COPD were given the former and all other patients received pentoxifylline. Evaluations were done comparing C-reactive protein (CRP) concentrations and ROX score (defined as the ratio of oxygen saturation measured by pulse oximetry/FiO2 to respiratory rate) between a control group constituted by COVID-19 patients receiving standard medication and the group receiving xanthines from day 1 to day 4 of therapy and results showed an increase in the ROX score (mean: 2.9) and a decrease in CRP (mean: −0.7) and mortality (24%) for the theophylline/pentoxifylline group. Even though it was a non-randomized study, the data obtained point out that this treatment could be associated with benefits for COVID-19 patients and warrants further research [16].

Additionally, Dahiya et al., reported the effects of theophylline or etophylline on sinus bradycardia, one of the most common arrythmias found in COVID-19 patients. This symptom is possibly related to viral myocarditis, myocardial ischemia or might be a side effect of COVID-19 medication. Ten COVID-19 patients that developed sinus node dysfunction received etophylline or theophylline prolonged release tablet (150 mg) once a day and a normal heart rate was monitored 72 hours after the treatment´s initiation. Even though the studied population was small, it seems that, if COVID-19 patients develop sinus bradycardia, a short lapse administration of either of these might be an effective treatment [18]. 

  • Wall, G. C., Smith, H. L., Trump, M. W., Mohr, J. D., DuMontier, S. P., Sabates, B. L., Ganapathiraju, I., Kable, T. J. Pentoxifylline or theophylline use in hospitalized COVID‐19 patients requiring oxygen support. Clin Respir J. 2021, 15(7), pp. 843-846. doi: 10.1111/crj.13363.
  • Dahiya, A.; Sharma, R.; Singh, A.; Joshi, P.; Wardhan, H. Role of Etophylline and Theophylline Prolonged Release Tablet in COVID-19 Associated Sinus Node Dysfunction. J Assoc Physicians India. 2022, 70(1), pp. 11–12.

Q= The authors should discuss and cite the already published paper of Rolta et al. "Methylxanthines as Potential Inhibitor of SARS CoV 2: an In Silico Approach." Current Pharmacology Reports 2022 Mar 8: 1-22

A= Thank you for your comments, we have added the reference and the following lines were added to section 4.3. Theophylline affinity with proteins Spike and Envelope (page 12, line 506-512):

Recently, in silico studies done by Rolta et al., showed interesting binding affinity between caffeine, methylxanthine, theobromine, and theophylline with SARSCoV-2 spike protein and S1 receptor-binding domain (S1RBD). Theoretically, all the above mentioned phytocompounds could potentially prevent the binding of SARS-CoV-2 to the ACE2 receptor sensibly diminishing the virus infectious capacity. Notwithstanding these promising results, these methylxanthine derivatives must be probed in vitro and in vivo to confirm their therapeutic benefits against SARS-CoV-2 [105].

  • Rolta, R.; Salaria, D.; Sharma, B.; Awofisayo, O.; Fadare, O. A.; Sharma, S.; Patel, C. N.; Kumar, V.; Sourirajan, A.; Baumler, D. J.; Dev, K. Methylxanthines as Potential Inhibitor of SARS-CoV-2: an In Silico Approach. Curr Pharmacol Rep. 2022, 8, 1-22. doi: 10.1007/s40495-021-00276-3.

Q= The authors also describe their own results in "4.3. Theophylline affinity with proteins Spike and Envelope", however only results are presented without any method described

A= Thank you for pointing out our mistake. We have added a supplementary file, which includes the following text:

Methodology

Molecular modeling of the E protein from SARS-CoV-2

Based on the primary sequence YP_009724392 of the E protein from SARS-CoV-2, it was possible to build the E protein of this virus by employing the crystal structure PDB: 5X29, which corresponds to the E protein of SARS-CoV. The identity percentage between the sequence of the E protein of SARS-CoV-2 and the crystal structure is 91.379%. A three-dimensional model of the pentamer of the E protein of SARS-CoV-2 was built by using Modeller 10.1 Software [1] and the three-dimensional structure of the Spike protein of SARS-CoV-2 Crystal 7WK3 was downloaded from the pdb database [2], which is the Omicron variant SARS-CoV-2 Spike protein crystal. The latter was chosen for the Docking studies.

Molecular docking

For the molecular docking of theophylline with both proteins (Spike and E of SARS-CoV-2) docking was performed directed at the RBD site (Spike protein) and the ion channel (E protein), respectively. The Vina program was used, with the following parameters: num_modes = 20, energy_range = 6 and exhaustiveness= 25. For the case of the Spike protein: center_x = 210.0, center_y = 170.0, center_z = 270.0, size_x = 60.00, size_y = 60.00 and size_z = 60.00. And for protein E: center_x = -7.0, center_y = 1.0, center_z = -6.0, size_x = 35.00, size = 30.00 and size_z = 30.0 [3].

Table 2. Theophylline Simulation Conditions

Parameter

Theophylline

Lipophilicity (Log Units)

0.89

Binds To

Albumin

Fraction Unbound

0.44

Molecular Weight (g/mol)

180.17

Compound Type And Pka

Acid / 8.8

Solubilility At Ref-Ph (mg/L)

14300

Ref-pH

6.5

Metabolizing Enzymes

CYP1A2

Metabolizing Enzymes Clearence (1/Min)

0.00843

Renal Clearence

0.15

Administration Protocol

Simple protocol

Administration Type

Intravenous bolus

Dose (mg/Kg)

1 and 10

Dosing Interval

Single

Table 3. Individuals Simulation Conditions

References.

Healthy

Renal impaired

Population

Mexican American – White (NHAES, 1997)

Mexican American – White (NHAES, 1997)

Age (Years)

70

70

Gender

Male

Male

Weight (Kg)

74.43

74.43

Height (cm)

165.96

165.96

BMI (Kg/m2)

27.02

27.02

Body Surface Area (m2)

1.85

1.85

Hematocrit

0.46

0.33

EHC continuos fraction

0

1

GFR Specific (ml/min/100 g organ)

26.6

5

Metabolizing enzymes expression

CYP1A2

CYP1A2

 [1] B. Webb and A. Sali, "Comparative Protein Structure Modeling Using MODELLER," Current Protocols in Bioinformatics, vol. 54, no. 1, 2016-06-01 2016, doi: 10.1002/cpbi.3.

 [2] H. M. Berman, "The Protein Data Bank," Nucleic Acids Research, vol. 28, no. 1, pp. 235-242, 2000-01-01 2000, doi: 10.1093/nar/28.1.235.

 [3] O. Trott and A. J. Olson, "AutoDock Vina: Improving the speed and accuracy of docking with a new scoring function, efficient optimization, and multithreading," Journal of Computational Chemistry, pp. NA-NA, 2009-01-01 2009, doi: 10.1002/jcc.21334.

Q= The article No13 of Amin O. Elzupir is in fact published 2022, but not in 2020, as is stated in the presented paper. 

A= Thank you for the observation, we have corrected the reference.

Q= All the cited literature must be revised as there are mistakes in 43 of 108 cited papers.

A= Thank you for the observation, we have checked all the references and used the correct format stated by the journal.

Reviewer 2 Report

The present manuscript describes "Theophyllin; old drug in anew light, application in COVID-19 through in silico studies". Theophylline (3-methyxanthine) is an old drug that is used to prevent and treat shortness of breath, asthma, chronic bronchitis and some other lung diseases.  The authors have explored the possibility of using theophylline drug for COVID-19 treatment. It is important to understand how this compound could behave against such disease not only at pharmacodynamic but also as pharmacokinetic level. Authors have done the quickest drug discovery through different computational methods either from network pharmacology or from quantitative systems pharmacology approaches.            Authors found that theophylline seems to be a relevant candidate for the treatment of COVID-19 patients since it aims several immunological targets involved in the pathophysiology of the disease. Through various mechanisms theophylline down-regulates the overactivation of the inflammatory processes activated by SARS-CoV-2. Additionally, it might have antiviral potential that could hinder the pathogenesis of the virus. Theophylline, offering multiple advantages with a known clinical use and an affordable price, should not be overlooked in the search for potential treatments for COVID-19 patients. I recommend this manuscript to publish in International Journal of Molecular Sciences.                                                                                                                                                                                                                                                                                                                                                                                 

Author Response

REVIEWER 2

The present manuscript describes "Theophyllin; old drug in anew light, application in COVID-19 through in silico studies". Theophylline (3-methyxanthine) is an old drug that is used to prevent and treat shortness of breath, asthma, chronic bronchitis and some other lung diseases.  The authors have explored the possibility of using theophylline drug for COVID-19 treatment. It is important to understand how this compound could behave against such disease not only at pharmacodynamic but also as pharmacokinetic level. Authors have done the quickest drug discovery through different computational methods either from network pharmacology or from quantitative systems pharmacology approaches. Authors found that theophylline seems to be a relevant candidate for the treatment of COVID-19 patients since it aims several immunological targets involved in the pathophysiology of the disease. Through various mechanisms theophylline down-regulates the overactivation of the inflammatory processes activated by SARS-CoV-2. Additionally, it might have antiviral potential that could hinder the pathogenesis of the virus. Theophylline, offering multiple advantages with a known clinical use and an affordable price, should not be overlooked in the search for potential treatments for COVID-19 patients. I recommend this manuscript to publish in International Journal of Molecular Sciences.

A= Thank you for the generous comments, we are pleased to hear that the work was to your liking.

Round 2

Reviewer 1 Report

I accept the revised manuscript. 

Author Response

Thank you very much your comments and suggestions enriching our manuscript.